# Atroposelective synthesis of biaxial bridged eight-membered terphenyls via a Co/SPDO-catalyzed aerobic oxidative coupling/desymmetrization of phenols

Shuang-Hu Wang[1], Shi-Qiang Wei[1], Ye Zhang[1], Xiao-Ming Zhang [2],
Shu-Yu Zhang [1], Kun-Long Dai[1], Yong-Qiang Tu [1,2] ✉, Ka Lu[2] & Tong-Mei Ding[1]

Bridged chiral biaryls are axially chiral compounds with a medium-sized ring connecting the two arenes. Compared with plentiful methods for the enantioselective synthesis of biaryl compounds, synthetic approaches for this subclass of bridged atropisomers are limited. Here we show an atroposelective synthesis of 1,3-diaxial bridged eight-membered terphenyl atropisomers through an Co/SPDO (spirocyclic pyrrolidine oxazoline)-catalyzed aerobic oxidative coupling/desymmetrization reaction of prochiral phenols. This catalytic desymmetric process is enabled by combination of an earth-abundant Co(OAc)$_2$ and a unique SPDO ligand in the presence of DABCO (1,4-diaza[2.2.2] bicyclooctane). An array of diaxial bridged terphenyls embedded in an azocane can be accessed in high yields (up to 99%) with excellent enantio- (>99% ee) and diastereoselectivities (>20:1 dr).

Biaryl compounds with an axial chirality are valuable architectures that present in bioactive natural products and promising pharmaceuticals[1–3]. The axial frameworks of these biaryl atropisomers also serve as privileged scaffolds of important chiral catalysts and ligands[4,5]. It is because of the widely recognized functions of these enantioenriched molecules that a prevailing research focus has centered on the atropenantioselective synthesis of axially chiral biaryls[6–8]. Despite recent achievements that continuously enrich the research field, relatively less methodologies are developed for the construction of bridged biaryls, a subclass of these compounds that bear a tether to connect the two arenes and form a medium-sized ring[9,10]. As such bridged biaryls, especially ones with a 7, 8, or 9 membered aza-ring linkage, are core structures of bioactive agents from either natural or medicinally relevant molecules (Fig. 1a), developing effective catalytic enantioselective approaches for their assembly would be highly desirable. However, compared with other biaryls, construction of axially chiral bridged biaryls would encounter synthetic challenges

associated with not only stereogenic formation of aryl–aryl axis but also assembling the medium-sized ring with torsional strain[9,10]. Simultaneously tackling those geometrical factors in a highly efficient and stereoselective manner would be nontrivial from the aspect of asymmetric catalysis. In this regard, several inspiring strategies have been developed, most involving the construction of the medium-sized ring from an achiral biaryl precursor (Fig. 1b, top left) or through an aromatization reaction with concurrent formation of an axial chirality (Fig. 1b, top right)[11–19]. In comparison, intramolecular coupling of tethered aryls was less studied with only a notable example taking advantage of Pd-catalyzed aryl-aryl coupling (Fig. 1b, bottom left)[20,21]. Direct oxidative coupling to access axially chiral bridged biaryls, to the best of our knowledge, has not been reported (Fig. 1b, bottom right), and synthesis of such biaryls bearing multiple axial stereochemical elements is also unprecedented[22]. Compared with a number of studies on the enantioselective formation of a single axis, development of effective methods to access chiral diaxial atropisomers is still in its

[1]School of Chemistry and Chemical Engineering, Frontiers Science Center for Transformative Molecules, Shanghai Jiao Tong University, Shanghai 200240, China. [2]State Key Laboratory of Applied Organic Chemistry and College of Chemistry and Chemical Engineering, Lanzhou University, Lanzhou 730000, China. ✉e-mail: tuyq@sjtu.edu.cn

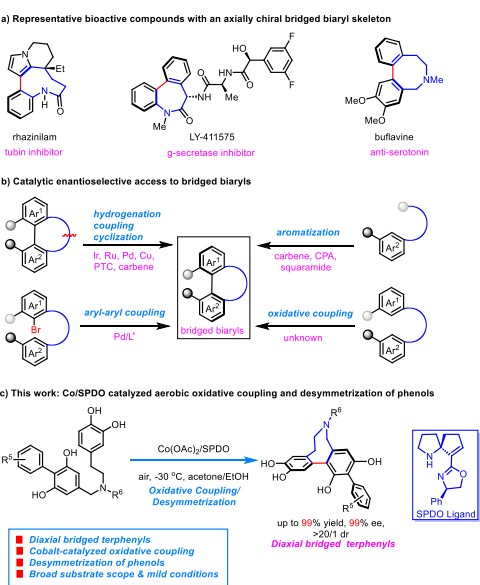

**Fig. 1 | Co/SPDO-catalyzed oxidative coupling/desymmetrization of phenols for synthesising chiral diaxial bridged biaryls. a** Representative bioactive compounds with an axially chiral bridged biaryl skeleton. **b** Catalytic enantioselective access to bridged biaryls. **c** Our design of Co/SPDO catalyzed aerobic oxidative coupling and desymmetrization of phenols.

infancy[22–34], probably due to the added challenges associated with controlling both diastereo- and atropenantioselectivities.

Considering the synthetic challenges of chiral diaxial bridged atropisomers and in continuation with our research interest in the synthesis of axially chiral compounds such as unsymmetrically substituted BINOLs and NOBINs via Cu/SPDO-catalyzed aerobic oxidative cross-coupling reactions[35,36], herein, we design and develop a Co/SPDO-catalyzed aerobic oxidative coupling/desymmetrization sequence of prochiral phenols for the enantioselective synthesis of biaxial bridged *m*-terphenyls embedded in an azocane (Fig. 1c). Notably, despite continued advances of earth-abundant metal-catalyzed oxidative coupling of phenols such as Cu, Co, Cr, Fe, or V[37], an asymmetric variant reaction catalyzed by cobalt, to the best of our knowledge, has not yet been achieved before our work[38,39].

## Results

### Reaction conditions optimization

To begin with, readily available dopamine derivative **1a** was chosen as a model substrate (Table 1). Since our previous studies have shown that variation of substitution or configuration of the oxazoline of SPDO ligand has a strong influence on its catalytic capability, we started by the preparation of a series of ligands, mainly by varying C4 substitution of oxazoline (L1-L4)[35,36,40]. Screening of these ligands was carried out under the catalysis of Co(OAc)₂ in acetonitrile at −30 °C with air as the terminal oxidant. Delightfully, the desired axially chiral terphenyl product **2a** was formed with excellent enantioselectivity after 58–80 h, and the reaction performed with phenyl-substituted ligand L1 would give the best result (entries 1-4). Notably, changing the C4 stereochemistry of oxazoline (L2-L4) would result in the complete reversion of the enantioselectivity, while modification of the pyrrolidine to a lactam would be detrimental to the reaction (entry 5). Cobalt salts were next screened (entries 6-8), and it turns out that inferior results were obtained when other cobalt salts were used either at −30 or 0 °C. Further investigation of solvents revealed acetone to be a better choice with respect to substrate solubility, product yield, and enantioselectivity. Next, we investigated some bases as the additive and found that addition of DABCO can significantly accelerate the

reaction rate (12 h) and improve both the diastereo- and enantioselectivity (entry 13). Finally, the yield and enantioselectivity could be further improved when the concentration of the reaction system was diluted using acetone/alcohol (3 mL/1 mL) as a mixed solvent (entry 15).

### Scope of sulfonyl substitution on nitrogen

With the above-optimized conditions identified, the scope of triaryl phenols was then explored (Fig. 2). Alternation of the sulfonyl protecting group was well-tolerated, and the reactions could deliver the desired products **2** in good yields with excellent enantioselectivities (92–95% ee) (**2a**–**2f**). Notably, the steric hindrance and electronic property of the sulfonyl benzene has a great influence on the diastereoselectivity (5/1–10/1 dr), with bulkier and electron-rich aryl group giving better results (**2c**–**2f**).

### Scope of aryl, heteroaryl, and naphthyl substitution

Next, we examined substrates with diverse substituents on the phenyl group (Fig. 3). A series of mono and polysubstituted phenyls were well-tolerated. The reactions took place smoothly to form the desired products in good yields with excellent dr and ee values regardless of either electron-donating or withdrawing substituent on arene (**2g-2w**). Interestingly, introduction of heteroatom and bulkier substituent on *ortho* position of the aryl group significantly improved the diastereoselectivity of the reaction (**2g-2n**). while strong electron-withdrawing substituents such as nitro-, cyano-, and carbonyl groups (**2u, 2w, 2ab**) would be detrimental to the diastereoselectivity. In addition, a range of heteroarenes that widely present in functional organic molecules, such as benzofuran, thianthrene, 9-fluorenone, dibenzofuran, and quinoline, were also amenable, affording products in 82–99% yields with 78–96% ee (**2x-2ac**). It is worth noting that a methyl-substituted substrate (**2ad**) is also capable of producing the expected product, albeit with only moderate enantioselectivity. The absolute configurations of **2ad** and **3a** (derivatization from **2a**) were assigned by single crystal X-ray diffraction analysis.

To further demonstrate the generality of the method, we studied triaryls with a naphthyl group (Fig. 3). In general, all substrates gave the desired products in good yields with good to excellent diastereo- and enantioselectivities. Similar to the above results, the introduction of polar functional groups on the *ortho* position of naphthalene could evidently improve the diastereoselectivity of the reaction (**2af, 2ah**), while a C4-substituent on naphthyl (**2ai-2ak, 2am-2ao**) would greatly reduce the diastereoselectivity. Substrates containing acenaphthene, fluoranthene, phenanthrene, and benzo[*b*]naphtho[1,2-*d*]thiophene moieties were also assessed, and the corresponding products were obtained with high yields and enantioselectivities (**2am-2ap**). Finally, for structural diversity, biaryls with a longer tether or a carbon linkage instead of a nitrogen were also prepared and investigated. However, these substrates (**1au-1ax**) failed to provide the coupling products (please see Supplementary Fig. 16 for details).

### Preparation of bridged atropisomers with a single stereogenic axis

Furthermore, a range of bridged eight-membered biaryl atropisomers with a single axial chirality was also interrogated (Fig. 4). Generally, good reaction outcomes were obtained with respect to yields and enantioselectivities. Notably, methyl protection of one of the hydroxyl group of resorcinol or replacement of *N*-sulfonyl protection with benzyl was allowed. The corresponding products were obtained with similar yields and enantioselectivities (**2as, 2at**). In addition, swapping the position of resorcinol and catechol group does not affect the viability of substrate. In this case, **2az** was obtained in high yield and enantioselectivity.

## Table 1 | Optimization of Conditions with 1a.[a]

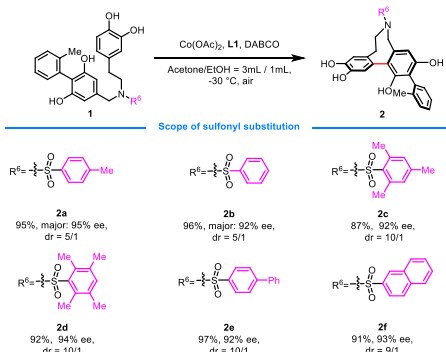

| entry | ligand | [Co] | solvent | time/h | yield (%)[b] | dr[c] | ee(%)[d] |
|---|---|---|---|---|---|---|---|
| 1 | L1 | Co(OAc)$_2$ | CH$_3$CN | 58 | 90 | 2.2/1 | 91 |
| 2 | L2 | Co(OAc)$_2$ | CH$_3$CN | 68 | 89 | 2.2/1 | -45 |
| 3 | L3 | Co(OAc)$_2$ | CH$_3$CN | 80 | 94 | 2.3/1 | -65 |
| 4 | L4 | Co(OAc)$_2$ | CH$_3$CN | 58 | 90 | 2.5/1 | -90 |
| 5[e] | L5 | Co(OAc)$_2$ | CH$_3$CN | - | N.R.[j] | - | - |
| 6 | L1 | CoCl$_2$ | CH$_3$CN | - | trace | - | - |
| 7[e] | L1 | CoCl$_2$ | CH$_3$CN | 12 | 95 | 3.5/1 | 74 |
| 8[e] | L1 | CoI$_2$ | CH$_3$CN | - | trace | - | - |
| 9 | L1 | Co(OAc)$_2$ | EtOH | 60 | 85 | 1.6/1 | 93 |
| 10 | L1 | Co(OAc)$_2$ | Acetone | 60 | 93 | 1.5/1 | 90 |
| 11[f] | L1 | Co(OAc)$_2$ | Acetone | 50 | 82 | 3.5/1 | 89 |
| 12[g] | L1 | Co(OAc)$_2$ | Acetone | 19 | 95 | 4.5/1 | 90 |
| 13[h] | L1 | Co(OAc)$_2$ | Acetone | 12 | 90 | 5/1 | 93 |
| 14[h] | L1 | Co(OAc)$_2$ | Acetone/EtOH=3/1 | 5 | 91 | 5/1 | 94 |
| 15[h, i] | L1 | Co(OAc)$_2$ | Acetone/EtOH=3/1 | 5 | 95 | 5/1 | 95 |

[a]The reaction was conducted with **1a** (0.1 mmol), Co salt (10 mol%), and ligand (12 mol%) in solvent (1 mL).
[b]Yield of the isolated product.
[c]Determined by $^1$H NMR analysis.
[d]The ee values were determined by UPC$^2$ or HPLC.
[e]At 0 °C.
[f]Pyridine (40 mol%) was added.
[g]Et$_3$N (40 mol%) was added.
[h]DABCO (1,4-diaza[2.2.2]bicyclooctane, 40 mol%) was added.
[i]Solvent (4 mL).
[j]N.R.: no reaction.

Fig. 2 | **Scope of sulfonyl substitution on nitrogen**[a]. [a]Reaction conditions: unless otherwise noted, the reactions were conducted with **1** (0.1 mmol), Co(OAc)$_2$ (10 mol %), L1 (12 mol%), and DABCO (40 mol%), in acetone/EtOH (3 mL/1 mL) at −30 °C. Isolated yields. The d.r. was determined by $^1$H NMR analysis of the crude mixture of products. The ee values were determined by UPC$^2$ or HPLC.

## Synthetic applications

The oxidative coupling of phenol **1a** could be carried out in gram-scale under the catalysis of 5% Co(II)/SPDO (Fig. 5). A slight decrease of the product yield and enantioselectivity was observed with maintained diastereoselectivity. Silyl protection of **2a** as its TBS ether was also performed to provide compound **3a** (please see Supplementary Fig. 17 for details).

## Proposed reaction mechanism

On the basis of previous literature reports and some mechanistic studies (please see Supplementary Figs. 12–15 for details), we proposed a possible reaction mechanism as depicted in Fig. 6. Specifically, the catalytic cycle begins with the formation of cobalt(III)/SPDO−superoxide species I[39,41,42]. Abstraction of a hydrogen atom (HAT) and an ensuing proton-coupled-electron transfer (PCET) process from catechol by cobalt-superoxide complex I generated a highly active *o*-benzoquinone II and a cobalt(II)/SPDO/H$_2$O$_2$ species[39,41], completing the oxidation process in the catalytic cycle. The following base-assisted 8-exo-trig cycloaddition catalyzed by cobalt(II)/SPDO serving as a Lewis acid proceeds from the *Re* face of the *o*-benzoquinone that is not shielded by the phenyl group of ligand and thus affords intermediate III with both central and axial chiralities[40,43,44]. Finally, the desired product **2** was produced after aromatization that renders a chirality transfer from central to axial, completing the catalytic process.

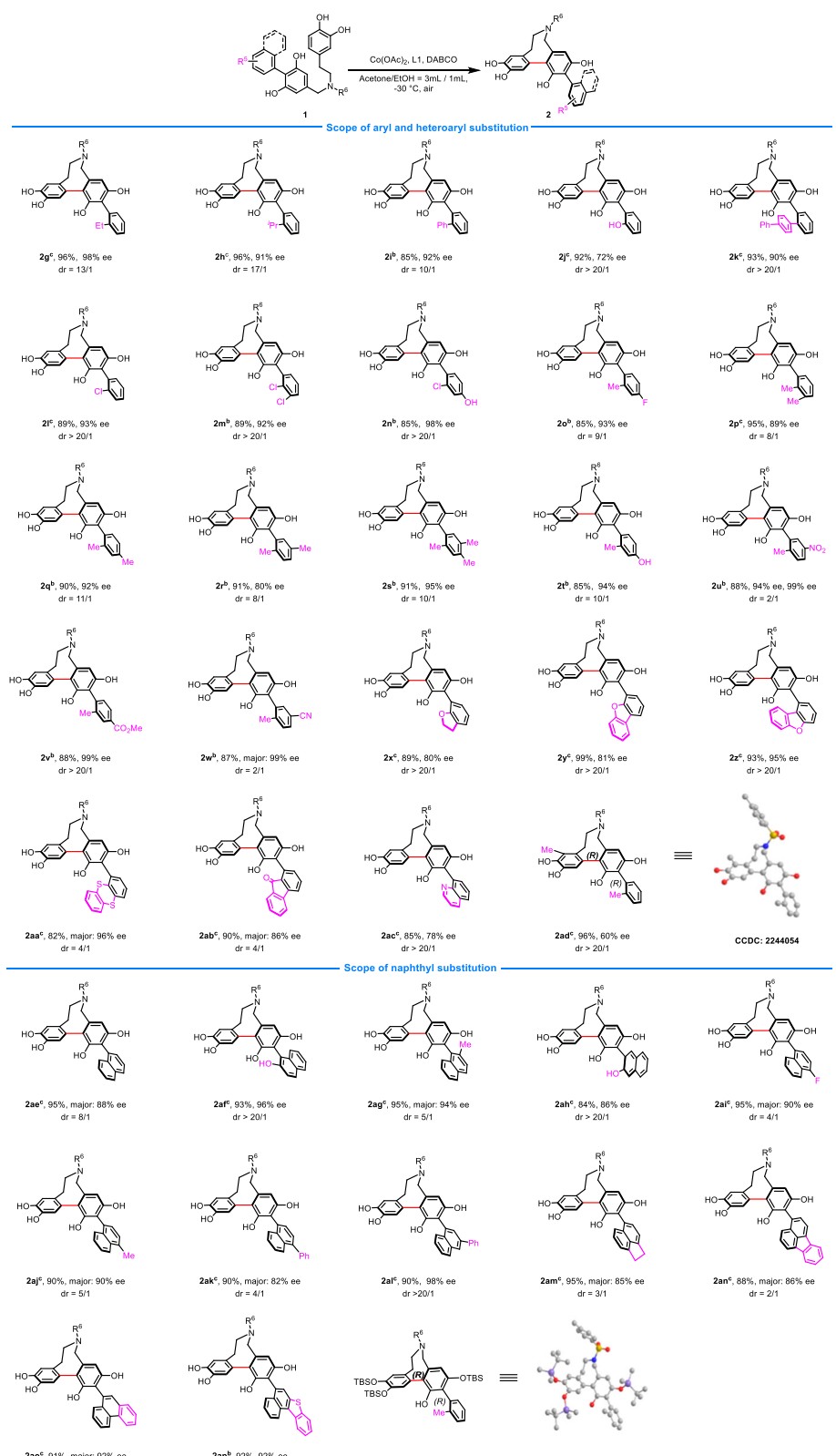

**Fig. 3 | Scope of aryl and naphthyl substitution[a].** [a]Reaction conditions: unless otherwise noted, the reactions were conducted with **1** (0.1 mmol), Co(OAc)$_2$ (10 mol %), L1 (12 mol%), and DABCO (40 mol%) in acetone/EtOH (3 mL/1 mL) at −30 °C. [b]R[6] = Mesitylene-2-sulfonyl. [c]R[6] = Tosyl. Isolated yields. The d.r. was determined by [1]H NMR analysis of the crude mixture of products. The ee values were determined by UPC[2] or HPLC.

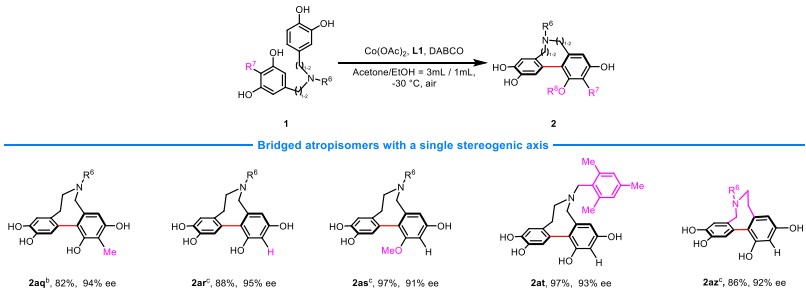

**Fig. 4 | Preparation of bridged atropisomers with a single stereogenic axis[a].**
[a]Reaction conditions: unless otherwise noted, the reactions were conducted with **1** (0.1 mmol), Co(OAc)$_2$ (10 mol%), L1 (12 mol%), and DABCO (40 mol%), in acetone/ EtOH (3 mL/1 mL), at −30 °C. [b]R$^6$ = Mesitylene-2-sulfonyl. [c]R$^6$ = Tosyl. Isolated yields. The ee values were determined by UPC$^2$ or HPLC.

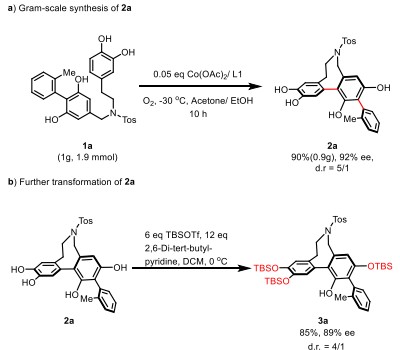

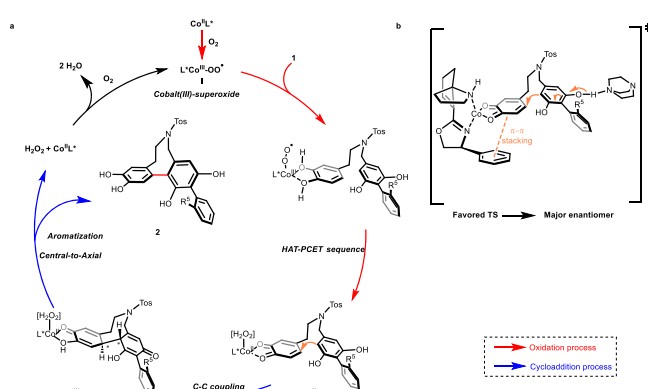

**Fig. 5 | Gram-scale reaction and derivatization. a** Gram-scale synthesis of **2a**. **b** Further transformation of **2a**.

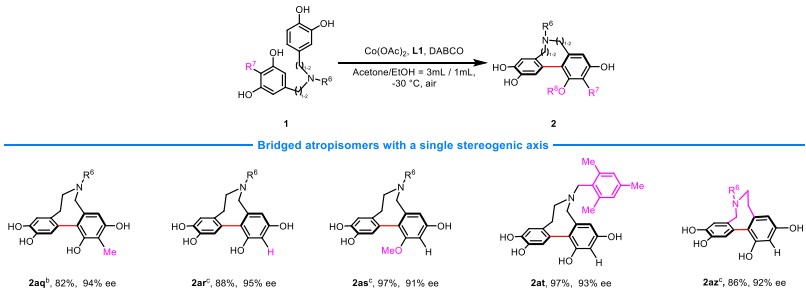

**Fig. 6 | Proposed reaction mechanism. a** The oxidation and cycloaddition process in the catalytic cycle. **b** Favored transition state.

## Discussion

In summary, we have successfully developed a novel method for the atroposelective synthesis of unprecedented biaxial bridged eight-membered terphenyl atropisomers through the Co(II)/SPDO-catalyzed aerobic oxidative coupling/desymmetrization sequence of phenols, featuring a broad substrate scope, good yields, excellent stereo-selectivities, and environmentally benign conditions. This work also represents the first asymmetric oxidative coupling of phenols with earth-abundant cobalt catalyst. Further applications of this catalytic system in other conversions are currently ongoing in our laboratory.

## Methods
### General information
All reactions were performed using oven-dried or flame-dried glassware equipped with a magnetic stir bar before used. All reagents were purchased from commercial suppliers and used without further purification. All solvents were purified by standard method. Toluene and tetrahydrofuran were distilled from sodium; dichloromethane (DCM) was distilled from calcium hydride; acetone and ethanol were purchased from commercial suppliers and used without further purification. Thin-layer chromatography was performed with EMD silica gel 60 F254 plates eluting with solvents indicated, visualized by a 254 nm UV lamp and stained with phos-phomolybdic acid. $^1$H NMR, $^{13}$C NMR and $^{19}$F NMR spectra were obtained on Bruker AM-400, Bruker AM-500. Chemical shifts ($\delta$) were quoted in ppm relative to tetramethylsilane or residual solvent as internal standard CDCl$_3$: 7.26 ppm for $^1$H NMR, 77.0 ppm for $^{13}$C NMR, D$_4$-CD$_3$OD: 3.31 ppm for $^1$H NMR, 49.00 ppm for $^{13}$C NMR; D$_6$-Acetone: 2.05 ppm for $^1$H NMR, 206.68 ppm and 29.92 ppm for $^{13}$C NMR; multiplicities are as indicated: s = singlet, d = doublet, t = triplet, q = quartet, m = multiplet. High-resolution mass spectral analysis (HRMS) data was measured on a Bruker impact II (Q-TOF) mass spectrum by means of the ESI technique. Crystallographic data were obtained from a Bruker D8 VENTURE diffractometer. Optical rotations were detected on RUDOLPH A21202-J APTV/GW. The enantiomeric excesses (ee) of the products were determined by high-performance liquid chromatography (HPLC) analysis or UPC$^2$.

### General procedure for the oxidative coupling/desymmetriza-tion reaction of prochiral phenols
Unless otherwise noted, reactions were performed: Co(OAc)$_2$ (1.7 mg, 10 mol%) and L1 (3.2 mg, 12 mol%) were added to acetone (3.0 mL) at room temperature, and stirred for 10 min. Then DABCO (4.48 mg, 40 mol%) was added and stirred for 1.5 h, then 1 mL EtOH was added to the system. The system was placed in a bath of ethanol at −30 °C 10 min later, and substrates (0.1 mmol) were added fol-lowed by stirring the reaction for 5–10 h. The reaction was quen-ched by addition of 40 µl HCl (2 mol/L in EtOAc), Then the product was purified by flash chromatography (DCM: Acetone = 8:1) on silica gel directly.

### Data availability
The data generated in this study are provided in the Supplementary Information file. The experimental procedures, data of NMR, and HRMS have been deposited in Supplementary Information file. Crystallographic data for the structures reported in this Article have been deposited at the Cambridge Crystallographic Data Centre, under deposition numbers CCDC 2244054 (**2ad**) and 2244056 (**3a**). These data can be obtained free of charge from The Cambridge Crystallographic Data Centre via www.ccdc.cam.ac.uk/data_request/cif." All other data are available in the main text or the Supplementary Information, or from the corresponding author upon request.

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

## Acknowledgements

We thank Prof. S.H. Hou (Shanghai Jiao Tong University) and B.M. Yang (Joint School of the National University of Singapore and Tianjin University) for valuable discussions. We thank Instrumental Analysis Center of SJTU for helpful spectral measurements. We acknowledge the NSFC (Nos. 21871117, YQT; 91956203, YQT; 22071147, YQT and 92256303, YQT), the "111" Program of MOE, Beijing National Laboratory for Molecular Sciences (BNLMS202109, YQT) for financial supports.

## Author contributions

The project was conceived and directed by Y.-Q.T., S.-H.W. designed the experiments and analyzed the data. S.-H.W., S.-Q.W., Y.Z., K.L., K.-L.D. and T.-M.D. performed the experiments. S.-H.W., X.-M.Z. and S.-Y.Z. prepared the manuscript. All authors discussed the results and commented on the manuscript.

## Competing interests

The authors declare no competing interests.
