## [Peer Review File · Nature Communications]

REVIEWER COMMENTS

Reviewer #1 (Remarks to the Author):

Recently, enantioselective synthesis of aromatics with multiple stereogenic elements has become a hot research topic of organic synthesis. However, effective synthetic methods have been rarely reported, owing to the formidable challenges in assembling multiple stereogenic elements with high enantiocontrol. This work by S. Wang and co-workers described a versatile method to achieve the atroposelective synthesis of bridged m-terphenyls with two C–C axes by a Co(II)/SPDO-catalyzed intramolecular oxidative coupling of phenols. A concurrent desymmetrization process enabled the construction of two C–C axial chiralities in a single step. The most eminent feature of this study is the use of earth-abundant cobalt catalyst and a unique spirocyclic pyrrolidine oxazoline (SPDO) ligand as the chiral source. Other features of this method include a broad substrate scope, good to excellent stereoselectivities, and environmentally benign conditions. Moreover, a reasonable reaction mechanism was proposed based on a few smartly designed control experiments and literature reports.

Overall, I treat this work a striking advance in multiple chiralities construction and Co-catalyzed asymmetric C–H bond activation. Thus, I recommend the publication of this manuscript in the esteemed *Nat. Commun.*, after addressing the following concerns.

(1) It is suggested to cite a few related references regarding multiple chiralities construction, e.g. *Nat. Commun.* 2021, 12, 4609, DOI: 10.1038/s41467-021-24678-5; *Angew. Chem. Int. Ed.* 2022, 61, e202208912; *Nature Synthesis*, 2022, 1, 709-718; *JACS Au* 2023, 3, 384–390.

(2) As revealed in Table 1, the SPDO ligands take a critical role in stereocontrol of the reaction. For example, the stereoselectivity can be totally reversed by L4 vs L1 (Entries 1 and 4, Table 1). How about the effect of other types of readily available ligands on this reaction? For example Salox ligands (see: *Angew. Chem. Int. Ed.* 2022, 61, e202208912; *Nature Synthesis*, 2022, 1, 709-718),

(3) The authors are encouraged to comment on the widely varied diastereomeric ratio (d.r.) of the products in Tables 3 and 4 (d.r. = 2:1 to >20:1).

(4) It's suggested to integrate the case 2at of Table 5 into Table 2 and change the corresponding expressions.

(5) It's suggested to integrate Table 3 and Table 4 into one Table and change the corresponding expressions.

(6) Line 81: "Reaction condition optimization" should be "Reaction conditions optimization"

(7) Lines 108 and 118: "Dr" should be "d.r."

(8) The expressions of Line 127-128 were superfluous.

(9) Line 160: "both central and axial chirality" should be "both central and axial chiralities"

(10) The absolute stereochemical configurations of products 2ad and 3a should be assigned based on the corresponding X-ray crystallographic analysis.

(11) The language of this manuscript should be polished. An incomplete list of improper expressions, e.g. "larger steric substituent"; "larger steric substituent"; "a longer chain length"; "maintained yields" should be corrected or modified.

The SI part

(12) "General procedures" should be "General procedure";

(13) The procedures regarding the preparation of racemic samples were missing.

(14) "Enantiomeric excess is 2a determined by HPLC" should be "Enantiomeric excess of 2a is determined by HPLC". Similar problems should be addressed throughout the SI.

(15) The style of chiral HPLC analysis for determining e.e. should keep consistent. Obvious differences were observed for 2b and 2c.

(16) Obvious differences between the retention time (Rt) of some racemic samples and the enantioenriched ones in chiral HPLC analysis were observed, e.g. 2d, 2g, 2h, 2l, 2o, 2p, 2t, 2x, 2y, 2ab, 2ad and 2ak.

(17) According to the reported integrations of chiral HPLC analysis, some of the racemic samples were not truly racemic, e.g. 2a, 2e, 2i, 2l, 2n, 2o, 2q, 2r, 2t, 2u, 2al. The authors should provide explanations on this issue.

(18) Diastereomeric ratio (d.r.) of the products can be determined by either ¹H NMR analysis of the crude mixture of products (details were not reported in SI) or chiral HPLC analysis, and the obtained d.r. value should be consistent. However, some of the d.r. values determined by racemic HPLC spectrum were significantly different from the ones determined by crude ¹H-NMR, e.g. 2u, 2w, 2aa, 2ag, 2ai, 2aj, 2ak, 2am, 2an and etc. The authors should provide explanations on this important issue.

(19) Only One decimal is required for the characterization of ¹⁹F-NMR (for 1ai and 2ai).

(20) The ¹⁹F-NMR of 1o and 2o were missing.

(21) For ¹H-NMR and ¹³C-NMR spectra characterization of F-containing samples (1o, 1ai, 2ai and 2o), the couplings between H-F and C-F were not properly reported.

(22) A few NMR spectra were not clean enough, and obviously extra peaks were found, e.g. 1x, 1aa, and 2as.

Reviewer #2 (Remarks to the Author):

The herein-submitted article reports a new methodology for the synthesis of bridged atropisomeric compounds bearing 2 chiral axes (a recent review published in Chem. Commun. on the synthesis of chiral compounds bearing 2 chiral elements should be cited). This topic is very timely and challenging, attracting the growing attention of the scientific community. The methodology developed by the authors implies intramolecular oxidative coupling of phenyl derivatives and is catalyzed by a chiral Co-catalyst which is certainly another strong point of the paper. Diastereoselectivities and enantioselectivities are generally high, while the products are isolated in good yields. However, the generality of this reaction is very limited, and a very specific substrate design is needed. Therefore the synthetic utility of such a coupling is limited. Could the authors also use less electron-rich compounds (no di-hydroxy-derived aromatics)? No post-functionalizations of the obtained compound are presented. No information is given about the rotational barriers and no mechanistic studies have been conducted. The authors propose a mechanistic cycle but it's based only on the literature precedents while stereoselectivity of this reaction cannot be rationalized and discussed. Besides, the HPLC data are rather poor (not publishable) for several compounds, such as 2e, 2h, 2m, 2s.

In conclusion, despite the timely character of the project and overall good results obtained, I think that this manuscript is for the moment too premature to be published in Nature Communications.

Reviewer #3 (Remarks to the Author):

Tu and co-workers reported a Co/SPDO-catalyzed aerobic intramolecular oxidative coupling/desymmetrization of phenols to afford biaxial bridged eight-membered terphenyls in excellent yields and ees and moderate to good dr. The stereochemical outcomes have been identified by X-ray diffraction. The phenols with different positions lead to the chemoselectivity. The resorcinol derivatives are used as nucleophilic partners. The catechol derivatives could coordinate with cobalt salt and be oxidized by air. This manuscript could be considered after addressing the following questions.

The various substrates (1au, 1av, 1aw, 1ax) have been synthesized in SI (P30-34), however, none of them showed in Scheme 11 worked. Based on the substrates, the linkers seem quite limited. What will happen during the exchange between resorcinol group and catechol group?

In scheme S10, the compound 1au is different from that in P30. It must be wrong.

In the title, the Co(II) has been emphasized. In the catalytic cycle, the Co(III) species has been proposed. Additionally, no direct evidence has been presented to support. The control experiments using Co(III) salts are suggested to carry out. This reviewer suggested to use cobalt instead of Co(II).

In the proposed mechanism, the π - π stacking effect has been used to explain the stereochemical outcome. However, in the table 1, the reaction using L2 with a phenyl group could deliver the desired

product with poor ee. There is similar π - π stacking effect using L2. The explanation should be reconsidered.

The deprotection reaction of the product with nitrogen should be conducted. It will be better to show the utility of the products.

Prof. Dr. Yong-Qiang Tu
School of Chemistry and Chemical Engineering,
Shanghai Jiao Tong University,
Shanghai 200240 P. R. China
Cell: +86-15002500060
E-mail: tuyq@sjtu.edu.cn

Nov. 23, 2023

Dear Reviewers:

According to your helpful comments and suggestions, we have carefully revised our original manuscript (NCOMMS-23-32554) and Supporting Information entitled “Atroposelective synthesis of biaxial bridged eight-membered terphenyls via a Co/SPDO-catalyzed aerobic oxidative coupling/desymmetrization of phenols”. Our detailed revisions and responses are listed below.

For Reviewer 1:

1. It is suggested to cite a few related references regarding multiple chiralities construction, e.g. Nat. Commun. 2021, 12, 4609, DOI: 10.1038/s41467-021-24678-5; Angew. Chem. Int. Ed. 2022, 61, e202208912; Nature Synthesis, 2022, 1, 709-718; JACS Au 2023, 3, 384–390.

Revision: Thanks for the reviewer's suggestions. We added the above references in our revised manuscript, as references 30-33.

2. As revealed in Table 1, the SPDO ligands take a critical role in stereocontrol of the reaction. For example, the stereoselectivity can be totally reversed by L4 vs L1 (Entries 1 and 4, Table 1). How about the effect of other types of readily available ligands on this reaction? For example Salox ligands (see: Angew. Chem. Int. Ed. 2022, 61, e202208912; Nature Synthesis, 2022, 1, 709-718).

Response: Thanks for the reviewer's suggestions. Corresponding product **2a** was obtained under the optimal conditions using Co(OAc)₂/**L6** as catalyst in 85% yield with 23% ee after 60 h. We also tried commercially available Co(II) and Co(III) catalysts (**Cat.6** and **Cat.7**) and a series of cobalt(III) salts, without obtaining the corresponding product. Catalyst **6** showed little catalytic activity with no formation of **2a** observed between -30 to 0 °C, while catalyst **7** would result in slow decomposition of substrate **1a**.

3. The authors are encouraged to comment on the widely varied diastereomeric ratio (d.r.) of the products in Tables 3 and 4 (d.r. = 2:1 to >20:1).

Response: The substituents on the aromatic ring were responsible for the dr ratio of the products. We added related comments on the dr ratio of product in our revised manuscript.

4. It's suggested to integrate the case **2at** of Table 5 into Table 2 and change the corresponding expressions.

Response: Thanks for the reviewer's suggestions. The product **2at** was a chiral compound with single axial chirality, which, we suppose, would be better in Table 5. Compounds in Table 2 contain biaxial chiralities.

5. It's suggested to integrate Table 3 and Table 4 into one Table and change the corresponding expressions.

Revision: In our revised manuscript, we have integrated Table 3 and Table 4 into one Table and have changed the corresponding expressions.

6. Line 81: "Reaction condition optimization" should be "Reaction conditions optimization"

Revision: In our revised manuscript, we have changed the "Reaction condition optimization" to "Reaction conditions optimization".

7. Lines 108 and 118: "Dr" should be "d.r."

Revision: In our revised manuscript, we have changed the "Dr" to "d.r."

8. The expressions of Line 127-128 were superfluous.

Revision: In our revised manuscript, we have deleted the sentence "Axially chiral molecules containing naphthalene widely exist in natural products, chiral catalysts, ligands and functional materials¹⁻³".

9. Line 160: "both central and axial chirality" should be "both central and axial chiralities".

Revision: In our revised manuscript, we have changed the “both central and axial chirality” to “both central and axial chiralities”.

10. The absolute stereochemical configurations of products **2ad** and **3a** should be assigned based on the corresponding X-ray crystallographic analysis.

Response: In our revised manuscript, we have assigned the absolute configuration of the product **2ad** and **3a**.

11. The language of this manuscript should be polished. An incomplete list of improper expressions, e.g. “larger steric substituent”; “larger steric substituent”; “a longer chain length”; “maintained yields” should be corrected or modified.

Revision: In our revised manuscript, we have changed the “larger steric substituent” to “bulkier substituent”, “a longer chain length” to “a longer tether”, “maintained yields” to “similar yields”.

The SI part

12. “General procedures” should be “General procedure”

Revision: In our revised Supporting Information, we have changed all the “General procedures” to “General procedure”.

13. The procedures regarding the preparation of racemic samples were missing.

Response: In our revised Supporting Information, the procedures for the preparation of racemic samples were added (please see **Scheme S9**).

14. “Enantiomeric excess is **2a** determined by HPLC” should be “Enantiomeric excess of **2a** is determined by HPLC”. Similar problems should be addressed throughout the SI.

Revision: In our revised Supporting Information, we have modified “Enantiomeric excess is xx determined by HPLC” to the “Enantiomeric excess of xx is determined by HPLC” and addressed other such problems.

15. The style of chiral HPLC analysis for determining e.e. should keep consistent. Obvious differences were observed for **2b** and **2c**.

Response: In our revised Supporting Information, we unified the style of chiral HPLC analysis. Due to the difficulties in separating the enantiomers of some of the compounds, two instruments, HPLC (n-hexane and isopropanol as eluent) and UPC² (carbon dioxide and methanol as eluent), were used to determine the e.e. values. Therefore, two different types of analysis data table were initially placed in SI.

16. Obvious differences between the retention time (Rt) of some racemic samples and the enantioenriched ones in chiral HPLC analysis were observed, e.g. **2d**, **2g**, **2h**, **2l**, **2o**, **2p**, **2t**, **2x**, **2y**, **2ab**, **2ad** and **2ak**.

Response: Thanks for the reviewer’s suggestions. The high polarity of polyhydroxy compounds may cause the obvious differences of retention time between racemic samples and the enantioenriched ones. Some literature precedence could be found. (J. Am. Chem. Soc.

2016, 138, 5202–5205; J. Am. Chem. Soc. 2015, 137, 15062–15065; Angew. Chem., Int. Ed. 2019, 58, 11023–11027; Angew. Chem., Int. Ed. 2020, 59, 2875–2880.). However, after extensive experimentation on HPLC, we could obtain the HPLC spectra with the similar retention times of both racemic and chiral samples including examples **2d**, **2g**, **2h**, **2l**, **2o**, **2p**, **2t**, **2x**, **2y**, **2ab**, **2ad** and **2ak**.

17. According to the reported integrations of chiral HPLC analysis, some of the racemic samples were not truly racemic, e.g. **2a**, **2e**, **2i**, **2l**, **2n**, **2o**, **2q**, **2r**, **2t**, **2u**, **2al**. The authors should provide explanations on this issue.

Revision: We have purified the racemic samples again (**2a**, **2e**, **2i**, **2l**, **2n**, **2o**, **2q**, **2r**, **2t**, **2u**, **2al**) and replaced the original HPLC spectra with suitable ones.

18. Diastereomeric ratio (d.r.) of the products can be determined by either ^1H NMR analysis of the crude mixture of products (details were not reported in SI) or chiral HPLC analysis, and the obtained d.r. value should be consistent. However, some of the d.r. values determined by racemic HPLC spectrum were significantly different from the ones determined by crude ^1H -NMR, e.g. **2u**, **2w**, **2aa**, **2ag**, **2ai**, **2aj**, **2ak**, **2am**, **2an** and etc. The authors should provide explanations on this important issue.

Response: Thanks for the reviewer's suggestions. We have recently found that the proportion of diastereomers of products could change after being placed for a period of time. Therefore, it is more accurate to measure the d.r. immediately after the reaction is complete. The ^1H NMR and HPLC data of crude products were directly recorded after the reaction within 1 hour, while the ^1H NMR and ^{13}C NMR data of purified products were given in our supporting information. During the period of purification, the diastereoselectivities of these products have changed, causing the problem of inconsistency. In our manuscript and supporting information, the diastereoselectivities (d.r. values) of crude products were given. Compounds **2al** and **2ar** are very stable at $-18\text{ }^\circ\text{C}$. However, at room temperature, the diastereoselectivity of **2al** would change from 24/1 to 1.8/1 in methanol solution during a period of 96 hours. Meanwhile, the enantioselectivity of the major product remains the same, and that of the minor product greatly enhances. These results indicate that one of the axes could rotate at ambient temperature, leading to an epimerization process. Next, we determined that the bridged axis was unstable by examining the stability of compound **2ar**. Finally, stability test of **2t** at different temperatures indicated that epimerization begins at around $16\text{ }^\circ\text{C}$. These results were also added in our revised Supporting Information.

2al, 90%, 98% ee in MeOH

entry	T ($^\circ\text{C}$)	time (h)	major ee (%) ^a	minor ee (%) ^a	dr ^a
1	25	0	98	-30	24/1
2	25	24	98	51	13/1
3	25	48	98	83	8/1

4	25	72	98	91	3/1
5	25	96	95	95	1.8/1
6	-18	30 days	98	-30	24/1

^aee and dr values were determined by HPLC analysis using a chiral stationary phase

2ar, 88%, 95% ee in MeOH

entry	T (°C)	Time (h)	Ee (%) ^a
1	25	0	95
2	25	24	88
3	25	48	87
4	25	120	78
5	-18	30 days	95

^aee values were determined by HPLC analysis using a chiral stationary phase

R = Mesitylene-2-sulfonyl

2t in MeOH

entry	T (°C)	Time (h)	Major ee (%) ^a	Minor ee (%) ^a	dr ^a
1	-18	12	94	-12	10/1
2	-13	12	94	-12	10/1
3	-8	12	94	-12	10/1
4	-6	12	94	-12	10/1
5	-2	12	94	-12	10/1
6	0	12	94	-12	10/1
7	2	12	94	-12	10/1
8	6	12	94	-12	10/1
9	8	12	94	-12	10/1
10	10	12	94	-12	10/1
11	12	12	94	-12	10/1
12	16	12	94	30	7/1
13	25	60	92	70	3.3/1

^aee values were determined by HPLC analysis using a chiral stationary phase

19. Only one decimal is required for the characterization of ¹⁹F-NMR (for 1ai and 2ai).

Revision: In our revised Supporting Information, we have modified the ¹⁹F-NMR of all fluorinated compounds.

20. The ¹⁹F-NMR of **1o** and **2o** were missing.

Response: In our revised Supporting Information, we have provided the ¹⁹F-NMR of all

fluorinated compounds in SI.

21. For $^1\text{H-NMR}$ and $^{13}\text{C-NMR}$ spectra characterization of F-containing samples (**1o**, **1ai**, **2ai** and **2o**), the couplings between H-F and C-F were not properly reported.

Revision: In our revised Supporting Information, we have modified the couplings between H-F and C-F for F-containing samples.

22. A few NMR spectra were not clean enough, and obviously extra peaks were found, e.g. **1x**, **1aa**, and **2as**.

Revision: In our revised Supporting Information, we have purified the sample of **1x**, **1aa**, and **2as**. The NMR spectra copies of **1x**, **1aa**, and **2as** have been replaced.

For Reviewer 2:

1. Could the authors also use less electron-rich compounds (no di-hydroxy-derived aromatics)?

Response: Thanks for the reviewer's suggestions. The nucleophilicity of compounds **1ar**, **1as**, **1ay** and **1ba** gradually decreases. The experimental results showed that no reaction occurs with **1ay** and **1ba** (please see **Scheme S12** in our revised supporting information). Therefore, the electron-rich compounds with at least one phenol hydroxy were required for a successful transformation.

2. No post-functionalizations of the obtained compound are presented.

Response: Thanks for the reviewer's suggestions. As shown in the following Scheme, we have conducted a gram-scale aerobic oxidative coupling of phenol **1a** using 5% Co(II)/SPDO catalyst and obtained compound **2a** with a similar yield and ee value. Next, we conducted some synthetic applications. First, we performed TBS protection of product **2a** and obtained compound **3a** with 85% yield and 89% ee. Second, we attempted to remove the protecting group on nitrogen of **2a** and **3a** but failed to obtain the corresponding products (**Scheme S14**).

a) Gram-scale synthesis of **2a**

b) TBS protection of substrate **2a**

c) Removing the protective group of substrate **2a, 3a**

3. No information is given about the rotational barriers

Response: Thanks for the reviewer's suggestions. Rotation of the bridged axis of some products was observed. A detailed discussion of the rotation phenomenon was placed after our response to question 18, reviewer 1.

4. No mechanistic studies have been conducted.

Response: The following mechanism process was proposed based on literature precedence and mechanistic studies.

The oxidation process was first confirmed. Autoxidation of Co(II)/SPDO under aerobic conditions is evident by UV-visible spectroscopy. Bubbling O₂ through an acetone/EtOH solution of Co(II)/SPDO resulted in an observed new absorption band, indicating oxidation of

Co(II)/SPDO (orange Line). Additionally, the resulting EPR spectrum reveals a signal of organic free radicals (Co(III)-superoxide). Next, abstraction of a hydrogen atom (HAT) and an ensuing proton-coupled-electron transfer (PCET) process from catechol by cobalt-superoxide complex **I** and generation of a highly active *o*-benzoquinone **II** was proposed based on known literatures.

Figure S1a. UV-visible spectra of Co/SPDO dissolved in acetone/EtOH at room temperature under N₂ (blue line), after bubbling O₂ through the solution (orange Line).

Figure S1b EPR spectra of Co/SPDO (1 mM) in O₂-saturated acetone/EtOH (3/1) at -100 °C.

Second, to further validate the proposed reaction mechanism, some control experiments were conducted. Monomethyl-protected substrate **1as** on resorcinol could deliver the desired product with good yield and enantioselectivity (**Scheme S12-A**). However, the dimethyl-protected substrate **1ay** couldn't undergo the cyclization process and offer the expected product due to the lower nucleophilicity (**Scheme S12-B**). On the basis of previous reports, silver oxide can be used to oxidize catechol instead of resorcinol to *o*-benzoquinone. In order to prove this process via the intermediate of *o*-benzoquinone, **1a** was first oxidized by silver oxide. As expected, the target product **2a** was obtained in 45% yield in the presence of 1 equivalent of silver oxide, which indicated that a highly active *o*-benzoquinone **II** was formed followed by 8-exo-trig cycloaddition to deliver intermediate **III** with both central and axial chiralities. Finally, the desired product **2** was produced after aromatization. It is worth noting that phenols cannot produce free radicals through silver oxide excluding the process involving radicals (**Scheme S12-C**). The process was also confirmed by free radical capture experiment. The addition of TEMPO only accelerated the reaction and could not prevent it

from proceeding (**Scheme S12-D**). In addition, to confirm that the catalytic cycle also involve *o*-benzoquinone intermediate **II**, we conducted an over oxidation experiment on compound **1a** and the over oxidation product **4a** (confirmed by **HRMS**) was obtained as expected, which indirectly proved that the process via the *o*-benzoquinone intermediate **II** (**Scheme S12-E**).

Scheme S12. Control experiments

Figure S2. HRMS-ESI for the over oxidation product **4a**

5. The HPLC data are rather poor for several compounds, such as **2e**, **2h**, **2m**, **2s**.

Response: In our revised Supporting Information, we have purified related compounds and redetermined the HPLC data of **2e**, **2h**, **2m**, **2s**.

For Reviewer 3:

1. The various substrates (**1au**, **1av**, **1aw**, **1ax**) have been synthesized in SI (P30-34), however, none of them showed in Scheme 11 worked. Based on the substrates, the linkers seem quite limited. What will happen during the exchange between resorcinol group and catechol group?

Response: Thanks for the reviewer's suggestions. We prepared compound **1az** with the position exchange of the resorcinol and catechol group (**Scheme S8**). Under the standard conditions, desired product **2az** could be obtained in 86% yield with high enantioselectivity (92% ee).

2. In scheme S10, the compound **1au** is different from that in P30. It must be wrong.

Revision: In our revised manuscript, we have numbered this compound as **1ay**.

3. In the title, the Co(II) has been emphasized. In the catalytic cycle, the Co(III) species has been proposed. Additionally, no direct evidence has been presented to support. The control experiments using Co(III) salts are suggested to carry out. This reviewer suggested to use cobalt instead of Co(II).

Response: In our revised manuscript, we modified Co(II) to Co in the title. We also tried a series of cobalt(III) salts and Co(III) catalyst without obtaining the corresponding product (please see our response to question 2 of review 1).

4. In the proposed mechanism, the π - π stacking effect has been used to explain the stereochemical outcome. However, in the table 1, the reaction using L2 with a phenyl group could deliver the desired product with poor ee. There is similar π - π stacking effect using L2. The explanation should be reconsidered.

Response: The facial selectivity we proposed during the 8-exo-trig cycloaddition was originated from the steric hindrance between the phenyl group of ligand and the nucleophilic resorcinol group, which results in the cycloaddition taking place from the *Re* face. The π - π stacking effect of *o*-benzoquinone and the phenyl group was suggested to enhance the steric influence through formation of a rigid structure, thus providing product with high enantioselectivity. Though a possible π - π stacking could also exist between *o*-benzoquinone and a benzyl group, we supposed that a “match-mismatch” effect might lead to a weakened stacking probably due to the flexibility of benzyl group as well as its spatial distance compared with the phenyl group. In such case, the cycloaddition might proceed with a lower facial selectivity.

REVIEWER COMMENTS

Reviewer #1 (Remarks to the Author):

The authors have addressed the main points raised by the reviewers to a good degree. I therefore recommend to accept this paper after the following minor concerns being addressed.

1. For Scheme S12-E, the yield of product 2a and 4a should be provided.
2. For Scheme S12-C, it is more reasonable to subject the isolated o-benzoquinone intermediate (oxidized by Ag₂O) to the standard reaction conditions to examine the formation of final product 2a.
3. For the deprotection of the N-Ts group, it is suggested to use the acidic conditions (see Sci. China Chem. 2023, 66, 3136) or Raney-nickel, EtOH (reflux) (see J. Am. Chem. Soc. 2014, 136, 4853).
4. It's suggested to perform DFT calculations to measure the rotational barriers of a representative product, and compare the calculated results with the observed experimental results.

Reviewer #3 (Remarks to the Author):

The revised manuscript is suitable for the acceptance.

Reviewer #4 (Remarks to the Author):

This review concerns only the crystallographic aspects of the manuscript and SI.

The Authors should be providing more details in the SI regarding the data collection, solution and refinement. These are industry-wide standards that are missing from the SI. All the authors have provided for details is "Crystallographic data were obtained from a Bruker D8 VENTURE diffractometer."

o Additional items such as instrument software (i.e. APEX 2, 3 4 or 5?), detector, solution method, refinement method, Hydrogen atom treatment, other refinement details, crystallization conditions should be included.

Within the manuscript authors state “The absolute configurations of 2ad and 3a (derivatization from 2a) were assigned by single crystal X-ray diffraction analysis.”

o Authors should specify what method was employed or state (further details in SI) then give the method there.

Comments on structure of 2ad.

- A re-refinement is necessary. They are missing the hydrogen atoms on the 4 alcohol groups.
 - Formula in the SI is missing the dichloromethane.
 - Small attention to detail items that are not going to change the conclusion however items that should be done are as follows.
- o Change list 6 to list 4.
- o Use a TWIN and BASF refinement.
- o Identify the structure solution method.
- o Will have to address (using a VRF-validation response form) the resulting B level alert due to Hydrogen bond donor not having an acceptor.

Comments on structure 3a structure

- The level B alerts need to be commented on using a VRF.
- Not sure what the solvent was that was used to crystallize 3a. This could provide some insight into the residual electron density that is causing the level B alert.
- Small attention to detail items that are not going to change the big picture conclusions made but should be done.

- o Change list 6 to list 4.
- o Use a TWIN and BASF refinement.
- o Identify the structure solution method.

Prof. Dr. Yong-Qiang Tu
School of Chemistry and Chemical Engineering,
Shanghai Jiao Tong University,
Shanghai 200240 P. R. China
Cell: +86-15002500060
E-mail: tuyq@sjtu.edu.cn

Mar. 26, 2024

Dear Reviewers:

According to your helpful comments and suggestions, we have carefully revised our original manuscript (NCOMMS-23-32554B) and Supporting Information entitled “Atroposelective synthesis of biaxial bridged eight-membered terphenyls via a Co/SPDO-catalyzed aerobic oxidative coupling/desymmetrization of phenols”. Our detailed revisions and responses are listed below.

For Reviewer 1:

1. For Scheme S12-E, the yield of product **2a** and **4a** should be provided.

Revision: Thanks for the reviewer's suggestions. We added the yield of **2a** in our revised Supporting Information. Due to the instability of *o*-benzoquinone **4a**, isolation of this compound failed. However, we obtained the **HMRS** data of **4a**.

2. For Scheme S12-C, it is more reasonable to subject the isolated *o*-benzoquinone intermediate (oxidized by Ag₂O) to the standard reaction conditions to examine the formation of final product **2a**.

Response: Thanks for the reviewer's suggestions. Due to the instability and high reactivity of the *o*-benzoquinone intermediate, this compound quickly converted into the final product **2a**. Unfortunately, we were unable to isolate the *o*-benzoquinone intermediate.

3. For the deprotection of the N-Ts group, it is suggested to use the acidic conditions (see Sci. China Chem. 2023, 66, 3136) or Raney-nickel, EtOH (reflux) (see J. Am. Chem. Soc. 2014, 136, 4853).

Response: Thanks for the reviewer's suggestions. When substrate **2a** was subjected to sulfuric acid conditions for 5 hours, no deprotection product was obtained with only a large amount of **2a** recovered. When substrate **2a** was subjected to Raney nickel in ethanol under reflux conditions, it was decomposed.

Unsuccessful attempts to remove the protecting group of substrate **2a**

4. It's suggested to perform DFT calculations to measure the rotational barriers of a representative product and compare the calculated results with the observed experimental results.

Response: Thanks for the reviewer's suggestions. In our original manuscript, the bridged chiral axis of compound **2ar** was found to rotate at room temperature, resulting the racemization. To further confirm that the bridged chiral axis in biaxial chiral products undergoes rotation at room temperature, the DFT calculations of rotation barrier of **2al** was performed. It is found that the rotation barrier of bridged chiral axis is 32.7 kcal/mol and another chiral axis is 42.0 kcal/mol, thus further indicating that the rotation of bridged chiral axis is possible which is consistent with our experimental results below.

2ar, 88%, 95% ee in MeOH

entry	T (°C)	time (h)	ee (%) ^a
1	25	0	95
2	25	24	88
3	25	48	87
4	25	120	78
5	-18	30 days	95

^aee values were determined by HPLC analysis using a chiral stationary phase

2al, 90%, 98% ee in MeOH

entry	T (°C)	time (h)	major ee (%) ^a	minor ee (%) ^a	dr ^a
1	25	0	98	-30	24/1
2	25	24	98	51	13/1
3	25	48	98	83	8/1
4	25	72	98	91	3/1

5	25	96	95	95	1.8/1
6	-18	30 days	98	-30	24/1

^aee and dr values were determined by HPLC analysis using a chiral stationary phase

For Reviewer 4:

1. The Authors should be providing more details in the SI regarding the data collection, solution and refinement. These are industry-wide standards that are missing from the SI. All the authors have provided for details is “Crystallographic data were obtained from a Bruker D8 VENTURE diffractometer.” Additional items such as instrument software (i.e. APEX 2, 3 4 or 5?), detector, solution method, refinement method, Hydrogen atom treatment, other refinement details, crystallization conditions should be included.

Revision: Thanks for the reviewer's suggestions. In our revised Supporting Information, we added more details about data collection, solution, and refinement.

2. Within the manuscript, authors state “The absolute configurations of **2ad** and **3a** (derivatization from **2a**) were assigned by single crystal X-ray diffraction analysis.” Authors should specify what method was employed or state (further details in SI) then give the method there.

Revision: Thanks for the reviewer's suggestions. In our revised Supporting Information, we added the state for determining absolute configurations of **2ad** and **3a**.

3. Comments on structure of **2ad**. A re-refinement is necessary. They are missing the hydrogen atoms on the 4 alcohol groups; Formula in the SI is missing the dichloromethane.

Revision: In our revised Supporting Information, the data of **2ad** has been re-refined, and the CIF file has been modified. Dichloromethane was added to the formula of **2ad**.

4. Small attention to detail items that are not going to change the conclusion however items that should be done are as follows. Change list 6 to list 4. Use a TWIN and BASF refinement. Identify the structure solution method.

Revision: We have updated the CIF file. In our revised Supporting Information, we added more details about data collection, solution, and refinement.

5. Will have to address (using a VRF-validation response form) the resulting B level alert due to Hydrogen bond donor not having an acceptor. Comments on structure **3a** structure. The level B alerts need to be commented on using a VRF.

Revision: We have updated the checkcif file.

6. Not sure what the solvent was that was used to crystalize **3a**. This could provide some insight into the residual electron density that is causing the level B alert.

Response: In our revised Supporting Information, we have added the crystallization solvent of compound **3a** and the solvent has been treated by squeeze.

REVIEWERS' COMMENTS

Reviewer #1 (Remarks to the Author):

The authors have addressed the main points raised by the reviewers. I therefore recommend to accept this paper.

Reviewer #4 (Remarks to the Author):

This review concerns only the crystallographic aspects of the manuscript and SI.

- Statement from the authors in the SI: The absolute configuration of compound 2ad, 3a is determined by the absolute structure parameter, with values of 0.032(13) and 0.027(17), respectively. If the absolute structure parameter are within ± 0.3 , it is generally considered that the absolute configuration has been

determined correctly.

- o This section needs to be revised as some of the terms are incorrect and the last statement is not true. Recommendations for papers relevant to assist the authors in resolving the issues and employing the correct terminology can be found here: DOI: 10.1107/S0021889800007184 and DOI: 10.1002/chir.20473

- o The values are fine and don't result in a different analysis, it's that the statements are wrong here.

- The details of the refinement are still missing. For example, there are no details about the use of a solvent mask (i.e. Squeeze). It is necessary to disclose its application in the SI including programs used and number of electrons the mask accounts for.

- It is also missing the necessary citations for the software and the programs used for example SHELXTL is not cited.

- There are missing information on what program were used for structure solution. It is stated that Intrinsic Phasing method was used, what program was employed? SHELXTL program package I don't believe has solution software based on this method.

- Treatment of hydrogen atoms attached to heteroatoms are generally placed and refined based on evaluation of the difference Fourier maps. This treatment of hydrogen atoms is not disclosed. It looks like the phenol group in 3a was placed using a riding model. Generally, this is disclosed or informed why the decision was made.

Still missing Value for `_atom_sites_solution_primary`

in both CIF VRFs a "hydrogen bond receptor" should probably be "hydrogen bond acceptor". Some would also argue that the acceptor here is the pi systems that is flanking the phenolic groups. Or that sterics imposed on the phenolic donor impede any traditional acceptor.

Overall these items are certainly not grounds for rejection and do not influence the final results. However they should be addressed before publication to meet standard expectations when reporting on X-ray crystal structures.

Prof. Dr. Yong-Qiang Tu
School of Chemistry and Chemical Engineering,
Shanghai Jiao Tong University,
Shanghai 200240 P. R. China
Cell: +86-15002500060
E-mail: tuyq@sjtu.edu.cn

May 1 2024

Dear Reviewers:

According to your helpful comments and suggestions, we have carefully revised our original manuscript (NCOMMS-23-32554C) and Supplementary Information entitled "Atroposelective synthesis of biaxial bridged eight-membered terphenyls via a Co/SPDO-catalyzed aerobic oxidative coupling/desymmetrization of phenols". Our detailed revisions and responses are listed below.

For Reviewer 4:

1. Statement from the authors in the SI: The absolute configuration of compound 2ad, 3a is determined by the absolute structure parameter, with values of 0.032(13) and 0.027(17), respectively. If the absolute structure parameters are within ± 0.3 , it is generally considered that the absolute configuration has been determined correctly. This section needs to be revised as some of the terms are incorrect and the last statement is not true. Recommendations for papers relevant to assist the authors in resolving the issues and employing the correct terminology can be found here: DOI: 10.1107/S0021889800007184 and DOI: 10.1002/chir.20473. The values are fine and don't result in a different analysis, it's that the statements are wrong here.

Revision: Thanks for the reviewer's suggestions, we have corrected the previous erroneous statement in Supplementary Information according to the reference DOI: 10.1002/chir.20473.

2. The details of the refinement are still missing. For example, there are no details about the use of a solvent mask (i.e. Squeeze). It is necessary to disclose its application in the SI including programs used and number of electrons the mask accounts for.

Revision: Thanks for the reviewer's suggestions. The details including solvent mask, programs used, number of electrons the mask accounts of the refinement have been added into Supplementary Information.

3. It is also missing the necessary citations for the software and the programs used for example SHELXTL is not cited.

Revision: Thanks for the reviewer's suggestions. We have added the citations for the software and the programs used in Supplementary Information.

4. There are missing information on what program were used for structure solution. It is stated that Intrinsic Phasing method was used, what program was employed? SHELXTL program package I don't believe has solution software based on this method.

Revision: Thanks for the reviewer's suggestions. We have added the missing information on what program were used for structure solution in Supplementary Information.

5. Treatment of hydrogen atoms attached to heteroatoms are generally placed and refined based on evaluation of the difference Fourier maps. This treatment of hydrogen atoms is not disclosed. It looks like the phenol group in 3a was placed using a riding model. Generally, this is disclosed or informed why the decision was made

Response: Thanks for the reviewer's suggestions. The H atoms attached to carbons were added geometrically and refined isotropically with the riding model. However, the H atoms of phenol group were added with Fourier maps.

6. Still missing Value for atom sites solution primary in both CIF VRFs a "hydrogen bond receptor" should probably be "hydrogen bond acceptor". Some would also argue that the acceptor here is the pi systems that is flanking the phenolic groups. Or that sterics imposed on the phenolic donor impede any traditional acceptor.

Revision: Thanks for the reviewer's suggestions. This value has been filled in the CIF file, and we have responded the CIF VRFs in the checkcif file.